# Parts per Million of Propanol and Arsine as Responsible for the Poisoning of the Propylene Polymerization Reaction

**DOI:** 10.3390/polym15173619

**Published:** 2023-09-01

**Authors:** Joaquín Hernández-Fernández, Rafael González-Cuello, Rodrigo Ortega-Toro

**Affiliations:** 1Chemistry Program, Department of Natural and Exact Sciences, San Pablo Campus, University of Cartagena, Cartagena 130015, Colombia; 2Chemical Engineering Program, School of Engineering, Universidad Tecnológica de Bolivar, Parque Industrial y Tecnológico Carlos Vélez Pombo, Km 1 Vía Turbaco, Turbaco 130001, Colombia; 3Department of Natural and Exact Science, Universidad de la Costa, Barranquilla 30300, Colombia; 4Food Packaging and Shelf-Life Research Group (FP&SL), Food Engineering Program, Universidad de Cartagena, Avenida del Consulado St. 30, Cartagena de Indias 130015, Colombia; rgonzalezc1@unicartagena.edu.co (R.G.-C.); rortegap1@unicartagena.edu.co (R.O.-T.)

**Keywords:** polypropylene, inhibitors, catalyst productivity, Melt Flow Index (MFI), computational chemistry, propanol, arsine

## Abstract

Polypropylene synthesis is a critical process in the plastics industry, where control of catalytic activity is essential to ensure the quality and performance of the final product. In this study, the effect of two inhibitors, propanol and arsine, on the properties of synthesized polypropylene was investigated. Experiments were conducted using a conventional catalyst to polymerize propylene, and different concentrations of propanol and arsine were incorporated into the process. The results revealed that the addition of propanol led to a significant decrease in the Melt Flow Index (MFI) of the resulting polypropylene. The reduction in the MFI was most notable at a concentration of 62.33 ppm propanol, suggesting that propanol acts as an effective inhibitor by slowing down the polymerization rate and thus reducing the fluidity of the molten polypropylene. On the other hand, introducing arsine as an inhibitor increased the MFI of polypropylene. The maximum increase in the MFI was observed at a concentration of 0.035 ppm arsine. This suggests that small amounts of arsine affect the MFI and Mw of the produced PP. Regarding the catalyst productivity, it was found that as the concentration of propanol in the sample increased (approximately seven ppm), there was a decrease in productivity from 45 TM/kg to 44 TM/kg. Starting from 10 ppm, productivity continued to decline, reaching its lowest point at 52 ppm, with only 35 MT/kg. In the case of arsine, changes in catalyst productivity were observed at lower concentrations than with propanol. Starting from about 0.006 ppm, productivity decreased, reaching 39 MT/kg at a concentration of 0.024 ppm and further decreasing to 36 TM/kg with 0.0036 ppm. Computational analysis supported the experimental findings, indicating that arsine adsorbs more stably to the catalyst with an energy of −60.8 Kcal/mol, compared to propanol (−46.17 Kcal/mol) and isobutyl (−33.13 Kcal/mol). Analyses of HOMO and LUMO orbitals, as well as reactivity descriptors, such as electronegativity, chemical potential, and nucleophilicity, shed light on the potential interactions and chemical reactions involving inhibitors. Generated maps of molecular electrostatic potential (MEP) illustrated the charge distribution within the studied molecules, further contributing to the understanding of their reactivity. The computational results supported the experimental findings and provided additional information on the molecular interactions between the inhibitors and the catalyst, shedding light on the possible modes of inhibition. Solubles in xylene values indicate that both propanol and arsine affect the polymer’s morphology, which may have significant implications for its properties and final applications.

## 1. Introduction

The Ziegler–Natta (ZN) catalyst, developed by Karl Ziegler and Giulio Natta in 1953, has been fundamental in the production of polymers such as polyethylene and polypropylene, which earned them the Nobel Prize in Chemistry in 1963 [1,2]. This catalyst has undergone modifications and improvements over time, becoming a key tool in the industry due to its ability to catalyze α-olefins and produce commercial polymers with various molecular structures [3]. Due to its wide industrial application, numerous studies have been carried out to continuously improve the efficiency and stereochemical specificity of the catalyst, focusing on finding more effective electron donors [4]. Some doublers generate essential changes in the stereospecificity of the catalyst by the selective inhibition of non-stereospecific sites [5,6,7,8]. The selective poisoning of the catalyst minimizes the number of available active centers and changes in the propagation rate of the polymer [9,10,11,12,13,14]. The reaction mechanisms that make it possible to explain the previously indicated interactions have not been fully elucidated and are still the subject of due debate [4]. In preparing highly active pre-catalysts for the production of polyolefins, a fundamental step is to introduce a monodentate Lewis base, such as alcohol, to activate the MgCl_2_ before its contact with TiCl_4_ [15]. Although attempts are made to remove most alcohol before polymerization, small residual amounts may still need to be entirely removed [16]. In addition, cross-contamination or adsorption on the surfaces and during the polymerization itself can occur, and side reactions that generate alcohol from the reagents used can occur; for example, if traces of water are present in the system, it can react with ether to form alcohol [17,18]. In this context, propanol has emerged as an inhibitor in the polymerization using the ZN catalyst [19]. Propanol acts as an electron donor and interacts with the catalyst and the active species, playing a crucial role as a modifying agent that regulates the catalytic activity and the structure of the resulting polymer [20,21,22,23]. It has been observed that propanol shows a preference for anchoring to the (110) surface of MgCl_2_ through chemical interactions, where the functional groups of propanol bind to the cations of Mg [9,19,23,24,25]. Instead, based on an examination of interaction energies between methanol, ethanol, and the ZN catalyst, it was revealed that these alcohols have a preference for forming more robust bonds with the five-coordinate Mg atom at the (101) surface than with the four-coordinate Mg atom on the (110) surface. The excellent stability of the 101 surface is related to the energetic stability provided by the hydrogen bonds between the chloride ion of the surface and the complexed alcohol [26].

It is essential to control and minimize the traces of propanol and other inhibitors during the polymerization process to avoid unwanted interference in the reaction [10]. One of the inhibitors that can also affect PP polymerization is arsine (AsH_3_) [27]. Arsine (AsH_3_) is an inorganic chemical compound of arsenic and hydrogen atoms [28]. Its structure has a central arsenic atom and three hydrogen atoms arranged in a tetrahedral shape. It can act as a soft Lewis base and bind transition metals that have partially filled as well as vacant s and p orbitals, forming M–ER3 complexes through covalent bonds between arsine and the metal [29]. In the synthesis of polypropylene (PP), the presence of arsine is relevant since, during the propylene polymerization stage, several essential components are used, such as propylene itself, the Ziegler–Natta (ZN) catalyst, the triethylaluminum co-catalyst (TEAL), selectivity regulators, hydrogen (H_2_) and nitrogen (N_2_) [30]. At least one of these components can act as a Lewis acid or contain transition metals in its structure. The interaction of arsine with these components can affect the catalytic process and the coordination of the metal in the ZN catalyst, as well as its interaction with the MgCl_2_ support that contains different active centers and the aluminum alkyl co-catalyst. It is essential to consider and control the presence of arsine and other inhibitors during polymerization to obtain a high-quality PP with the desired physicochemical properties [15,28,31,32,33,34].

This research studies propanol and arsine as inhibitors of the Ziegler–Natta catalyst in polypropylene production. Most research on polypropylene polymerization focuses on catalysts and reaction conditions; however, this research stands out by looking in detail at two specific inhibitors, propanol and arsine. Analyzing how these inhibitors affect the catalytic process and the properties of the resulting polymer is a novel and uncommon approach in the field. The study includes analyses of the essential properties of polypropylene, computational calculations, and reactivity profiles to profoundly and holistically understand how inhibitors affect polymerization. This study is aligned with the sustainable development goals and benefits the chemical industry in terms of efficiency and sustainability.

## 2. Materials and Methods

### 2.1. Reagents and Materials

The Ziegler–Natta/MgCl_2_ catalyst, co-catalyst of triethylaluminum 98%, and cyclohexyl methyl dimethoxysilane as the selectivity agent were purchased from Merck, Darmstadt, Germany. Propylene is used as a monomer and was obtained from Petroquímica, Cartagena, Colombia. Additionally, hydrogen and nitrogen, as well as arsine and propanol, with a purity of 99.999%, were used.

### 2.2. Preparation of the Pre-Catalyst

To prepare the pre-catalyst, the scientific literature was taken as a basis [27]. (1). The inhibitors were independently mixed with MgCl_2_ for 4 h. (2). The suspension was cooled. (3). It was washed with heptane. (4). The filtrate was eliminated, and the washing process was repeated six times until a white powder was obtained. (5). We added TiCl_4_ and heated the mixture for 2 h at 90 °C. (6). Then, we cooled and washed the crystals five times with toluene [27].

### 2.3. Polymerization Procedure for Propylene

The synthesis was carried out in a 1 L capacity autoclave, and the determination of raw materials and reaction conditions were carried out according to those described in the literature [35]. A suspension of TEAL/heptane pre-catalyst was prepared [35]. Hydrogen (30 g/h) and propylene (1.2 MT/h) were added [9,10,11,12,35]. The mixture was heated to 71 °C and stirred at 510 rpm at 27 bar. The suspension obtained from the PP was filtered, washed, and dried under a vacuum at 90 °C [9,10,11,12,35]. Samples of the final polypropylene obtained were taken, as described in Table 1 and Table 2, to observe the effect caused by these inhibitors (propanol and arsine) on properties such as the MFI, Mw, etc.

### 2.4. Computational Details

To carry out the computational calculations, the Gaussian 09, Revision D.01, and the theoretical method of the density functional theory (DFT) were used. The B3LYP functional method was chosen together with the 6-311G(d,p) base set to improve the structure of the inhibitors and the catalyst. The MgCl_2_ surface was modeled using the (110) termination, as this option resulted in a more favorable TiCl_4_ adsorption energy compared to other surfaces, such as (001) and (104) [36]. The MgCl_2_ solid is composed of two-dimensional layers of Cl-Mg-Cl, held together by van der Waals forces. These layers are perpendicular to the surface (110), and their terminations form one-dimensional rows. A model was used to simulate the reaction environment consisting of a 3 × 3 surface (110) supercell of MgCl_2_ containing three rows, each with 3 Mg atoms on the surface, and anchored by a TiCl_4_ unit.

Computer tools such as Gaussview were used to visualize the optimized structures and analyze the HOMO (highest level occupied) and LUMO (lowest unoccupied level) orbitals. The energy values of these orbitals were used to explore the global chemical reactivity properties, such as electronegativity (χ) and chemical potential (μ). Characteristics such as the overall softness (S), nucleophilicity index (N), electrophilicity index (ω), and hardness (η) were also examined. In addition, a molecular electrostatic potential (MEP) map was generated to visualize the electrical charge distribution of each of the compounds of interest. Likewise, the GaussSUM 3.0 software was used to use the UKA FOKUI function to calculate the values of the chemical descriptors and the local reactivity properties.

#### Interaction of Propanol and Arsine with the Active Site of Ti (Inhibitor–Ti Interaction)

The dynamic interaction between inhibitor adsorption and the titanium (Ti) center, as well as with isobutyl, has been analyzed through classical trajectory calculations using the B3LYP-D3/6-311G(d,p) methodology. All simulations were performed using the density functional theory (DFT) and Gaussian software 16, Revision C.01. To achieve accurate equilibrium geometries, geometry optimization was carried out under the B3LYP-D3 functional, as it has previously been confirmed to provide optimal geometry results in ZN catalytic systems. Electronic descriptions of the atoms were based on the triple-ξ basis sets, augmented with polarization functions (6-311G(d,p) Gaussian basis set).

After the geometry optimizations, single-point calculations were performed using the B3LYP/6-311G theoretical level on the geometries previously optimized under B3LYP-D3/6-311G. The choice of this methodology is supported by the previously demonstrated ability to achieve agreement with the experimental results when using B3LYP, which is widely employed in molecular simulations of transition metal-catalytic systems [37].

Adsorption simulations were based on a surface model using the Mg_14_Cl_28_ cluster derived from relaxed MgCl_2_ surfaces. The choice of this model is based on previous research indicating the weakness or even instability of TiCl_4_ coordination with the (104) plane, contrasting with the highly favorable energy of TiCl_4_ coordination with the (110) plane [37]. The adsorption energy of the inhibitor, $E_{ad}$, is calculated according to Equation (1):(1)$$E_{ad}=E_{Mg/p}-E_{Mg}-E_{P}$$

In this scenario, $E_{Mg/p}$ refers to the energy of the system composed of a toxic substance molecule attached to the surface of MgCl_2_. $E_{Mg}$ and $E_{P}$ are the energies of the uncovered MgCl_2_ system and the toxic substance in its free state, respectively.

### 2.5. Determination of the Fraction Soluble in Xylene of PP

The determination of the fraction soluble in xylene is a test that allows us to measure the proportion of polypropylene polymers that dissolve in xylene. This soluble fraction is mainly related to the amorphous portion present in polypropylene. Using xylene as a solvent is preferred in this test, as it is more specific for detecting the atactic part of the polymers compared to other solvents. It has been observed that the concentration of the soluble fraction obtained with xylene is closely correlated with the performance of the polymer in specific applications, such as the manufacture of films or fibers. In other words, the amount of soluble fraction can influence how the polymer performs in different end uses. In this investigation, the soluble fraction in xylene (XS) was measured at a temperature of 25 °C using the procedure established in the ISO 16152 standard. This measurement provides us with valuable information on the composition and characteristics of the polypropylene that is being studied [38].

### 2.6. Melt Flow Index (MFI)

A Tinius Olsen MP1200 plastometer was used to measure the polypropylene flow rate (MFI). The plastometer maintained a constant temperature of 250 °C in its cylinder and used a 2.40 kg piston to displace the polypropylene melt. After obtaining the MFI data, the average molecular weight of each polypropylene sample was evaluated using Equation (2), known as the Bremner approximation [39].
(2)$${\bar{M}}_{w}=-10332\times lnMFI+76829$$

## 3. Results

### 3.1. Interaction of Propanol with the Ziegler–Natta Catalyst

According to previous research, propanol, like the catalyst, coordinates on MgCl_2_ surfaces where the magnesium atoms do not have a fully occupied coordination sphere. These surfaces include the (100), (101), (104), and (110) planes. At the (100) surface, magnesium atoms can have three, five, or six coordinations. However, after reconstruction of the (100) surface, it is more likely that the magnesium atoms are coordinated at five positions. Also, on surfaces (101) and (104), the magnesium atoms are coordinated at five positions. The (104) surface is flat at the atomic level, as is the (110) surface, where the magnesium atoms are coordinated in four positions [26]. As mentioned above, we can assume that during the activation of the MgCl_2_ support, the alcohols coordinate on surfaces where the magnesium atoms have vacant coordination sites. Our study chose the (110) surface with four-position coordinated magnesium atoms as a typical example of this surface, as seen in Figure 1.

The analysis of the interaction energies shows that the alcohols form very stable complexes with the Mg_5_Cl_10_ group. However, the bond strength depends on the overlapping effects. The stability of the alcohol complexes in the Mg_5_Cl_10_ group varies according to the coordination number of the magnesium atom [26]. When alcohols coordinate with the four-coordinated magnesium atom on the surface (110), the stability of the complexes increases with the size of the alcohol [26]. At the (110) surface, the coordination angle between the Mg-O bond and the surface, depicted in Figure 2c, is nearly rectangular (84–89°) when an alcohol donor coordinates with the magnesium atom [26]. The coordination sphere of the oxygen atom is almost planar, partly due to a weak π bond between the magnesium atom and the oxygen atom [26]. This coordination geometry for alcohols is energetically favorable; in fact, the energy released when coordinating a second donor is only two-thirds of that released with the first donor [26].

### 3.2. Adsorption of Propanol Molecules on TiCl_4_-MgCl_2_

How a propanol molecule interacts with the uncoated surface of the support (MgCl_2_) was investigated using an extended cell. The computational results show that propanol binds to the oxygen atom by interacting with a magnesium atom on the surface in a Lewis base–Lewis acid interaction, as shown in Equation (3).
(3)propanol+surface→propanol−surface

Once TiCl_4_ adsorbs to the surface of MgCl_2_, the magnesium atoms on the surface are no longer equivalent. Therefore, we extended our study on the interaction of propanol with the entire catalyst surface, considering all possible different Mg interaction sites. Analyses of the global and local descriptors to the TiCl_4_ and the MgCl_2_ support were carried out using the density functional theory and the Fukui function to predict the relative reactivity at different sites of both structures. According to the calculations, the sites most prone to electrophilic, nucleophilic, and free radical attacks on the MgCl_2_ support were identified. The results indicate that atoms at positions 1, 3, 6, 9, 28, 30, and 37 show a higher susceptibility to nucleophilic attack; specifically, 28 and 30 represent a surface Mg atom close to TiCl_4_, which are much more prone to nucleophilic attack by propanol oxygen. On the other hand, position 13 is the prominent site for an electrophilic attack. Regarding free radical attacks, it was also found that the atoms in positions 1 and 13 are the most susceptible sites (see Figure 1). We also considered the direct interaction between propanol and TiCl_4_ (propanol–TiCl_4_) to complete the system description.

#### 3.2.1. Frontier Molecular Orbitals and Global Descriptors of Propanol, Arsine, TiCl_4_, and MgCl_2_

A theoretical study was carried out to analyze the molecular orbitals related to electron transfers, with the aim of understanding how propanol reacts chemically. The energy variation between the highest occupied orbital (HOMO) and the lowest unoccupied orbital (LUMO) in various possible electronic transitions was investigated. In addition, different global chemical reactivity properties were calculated to understand how the charge is transported in propanol. These properties included electronegativity (χ) and chemical potential (μ). Characteristics such as the overall softness (S), nucleophilicity index (N), electrophilicity index (ω), and hardness (η) were examined [40]. These calculations were based on the ionization potential and electron affinities associated with the lowest energy electron transfers [41]. In Figure 3a, a charge transfer can be observed in the molecule, where electrons move from the highest occupied orbital (HOMO) to the lowest unoccupied orbital (LUMO). This charge transfer causes a change in the distribution of electron densities within the molecule. Table 3 shows the values obtained for the chemical potential (μ = −2.5936), the global softness (S = 0.2221), and the nucleophilicity index (N = 2.0645), which indicate that propanol tends to donate electrons to other chemical species during a chemical reaction. This means the compound acts as an electron donor in the PP polymerization reaction.

The results obtained for arsine show that it has a high capacity to donate electrons to other chemical species that act as electron acceptors in a chemical reaction. This is because it has a negative chemical potential, indicating that it has energy available to donate electrons. Furthermore, its overall softness suggests that it can accept or donate electrons, which favors its reactivity as an electron donor. Its electronegativity value reveals a strong affinity for the electrons of other chemical species. On the other hand, the electrophilic index shows that arsine tends to accept electrons from other chemical species with a nucleophilic character; that is, they tend to donate electrons. These data indicate that arsine is highly prone to participate as an electron donor in chemical reactions (see Figure 3b and Table 3).

Figure 4a shows the HOMO (highest occupied orbital) and LUMO (lowest unoccupied orbital) orbitals of TiCl_4_, which are fundamental to understanding the chemical reactivity of the molecule. The difference in energy between the HOMO and the LUMO is known as the “energy gap.” A large gap, as in this case (ΔE = 4.9772 eV), suggests that the molecule has a wide gap between the energy level occupied by the electrons and the lowest energy level available to receive electrons. This indicates more excellent molecule stability since the energy required to excite electrons from the HOMO to the LUMO is significantly higher. The magnitude of the energy gap is relevant to the chemical reactivity of TiCl_4_. A large gap indicates that the molecule is less reactive and less likely to participate in chemical reactions with other species. On the other hand, if the energy gap were smaller, the molecule would be more prone to undergo reactions since the electrons could be more easily excited from the HOMO to the LUMO and vice versa. The Homo−1 and Lumo+1 orbitals, being the closest in energy to Homo and Lumo, respectively, can significantly influence the reactivity properties of the molecule. If their energies were very different from those of Homo and Lumo, they could affect the relative stability of the molecule and its ability to participate in chemical reactions. The energy difference between Homo−1 and Lumo+1 (ΔE) is also relevant because it measures the extent of the energy gap between the highest occupied and lowest unoccupied orbitals. A large ΔE indicates that the molecule has a significant energy gap between these orbitals, suggesting more excellent stability and a lower probability of chemical reactions. The Homo and Lumo orbitals are directly involved in many chemical reactions. Homo is usually associated with donating electrons, while Lumo is associated with accepting electrons. However, the Homo−1 and Lumo+1 orbitals can also participate in reactions, especially in the electron transfer processes. Its energy and distribution can influence the availability of electrons and, therefore, the molecule’s reactivity.

In the magnesium chloride support context, an intermolecular charge transfer is observed in Figure 4b, where electrons move from the highest occupied orbital (HOMO) to the lowest unoccupied orbital (LUMO). The values obtained for the chemical potential (−5.3813), the overall smoothness (0.4038), and the electrophilicity index (5.1161) indicate that the molecule is very likely to accept electrons during chemical reactions in the presence of a Lewis base or a nucleophile. This suggests magnesium chloride (MgCl_2_) powerfully acts as an electron acceptor. In addition, the electronegativity value (5.3813) reveals that MgCl_2_ shows a significant affinity towards other chemical species’ electrons, reinforcing its ability to accept electrons and participate in charge transfer reactions. The nucleophilic index (0.1954) indicates that the molecule has a relatively low probability of acting as a nucleophile in reactions with electrophiles. This suggests that MgCl_2_ is more likely to receive electrons than donate them when interacting with electrophiles. These values are reported in Table 4.

#### 3.2.2. Local Descriptors for Propanol, Arsine, TiCl_4_, and MgCl_2_

The Fukui function is an essential tool in theoretical and computational chemistry that allows us to predict and understand the chemical reactivity of molecules. This function, developed by the Japanese chemist Kenichi Fukui, winner of the Nobel Prize in Chemistry in 1981, is based on the electron density theory [42]. In quantum chemistry, calculation methods such as the DFT (density functional theory) are used to obtain the Fukui functions, which describe the chemical reactivity of the atoms in a molecule. These functions are expressed mathematically by equations such as (4)–(6), where $q_{r}$ represents the atomic charge and *N*, *N*_+1_, and *N*_−1_ indicate neutrality, an anionic site, and a cationic site, respectively. These equations allow us to evaluate and compare how atoms can react and participate in charge changes under different chemical conditions. The Fukui function provides valuable insight into how a molecule interacts with other chemical species and how electronic changes can occur in its atoms during a chemical reaction [43,44].
(4)$${f_{r}}^{0}=\frac{1}{2}\left[q_{r}\left(N_{+1}\right)-q_{r}\left(N_{-1}\right)\right]$$
(5)$${f_{r}}^{-}=q_{r}\left(N\right)-q_{r}\left(N_{-1}\right)$$
(6)$${f_{r}}^{+}=q_{r}\left(N_{+1}\right)-q_{r}\left(N\right)$$

The Fukui function is divided into two parts: the nucleophilic Fukui function, *f_r_*^+^, which indicates the areas that are most likely to be attacked by nucleophilic species, and the electrophilic Fukui function, *f_r_*, which shows the areas that are most likely to be attacked by electrophilic species. These functions help identify the reactive sites in a molecule. Furthermore, the dual descriptor Δ*f_r_* is used, which is the difference between *f_r_*^+^ and *f_r_*^−^. If Δ*f_r_* is positive, it indicates a preference for a nucleophilic attack in those areas. On the other hand, if Δ*f_r_* is negative, it means a preference for an electrophilic attack in those areas. To better understand how a molecule reacts, we performed calculations and created tables showing the Fukui functions (ƒ^0^, ƒ^+^, and ƒ^−^) for each site in the molecule. These results provide us with information about the qualitative reactivity and selectivity of specific sites within the molecule. Table 5 shows the particular values of the Fukui functions and the dual descriptor for each location in the propanol molecule. This table allows us to identify which areas of the molecule are most likely to react and how an interaction with other chemical species can occur. The same calculations were applied for arsine, TiCl_4_, and the support (see Table 6, Table 7 and Table 8).

The calculations have made it possible to identify the sites most susceptible to different types of chemical attacks in propanol. According to the results, the oxygen atom at position 11 and the hydrogen atoms at positions 10 and 9 are the most prone to electrophilic attack. On the other hand, the atoms most susceptible to nucleophilic attack are found at positions 1, 8, 9, 10, and 11. Furthermore, it was found that carbon at positions 1 and 8, oxygen at position 11, and hydrogen atoms at positions 9 and 10 are the sites most prone to free radical attack (Figure 5).

These findings are supported by Table 5, which shows the variations in the dual descriptor (Δ*f*) about atoms. The results indicate that the oxygen atom at position 11 has a negative value of Δ*f* (−0.2748), suggesting it is more prone to electrophilic attack. On the other hand, the carbon at position 8 has a positive value of Δ*f* (0.1798), indicating that it is a more favorable site for a nucleophilic attack.

In arsine’s case, the calculations revealed that the arsenic atom shows greater reactivity and is the site most susceptible to electrophilic attack. On the other hand, hydrogen atoms are the points most vulnerable to nucleophilic attack, as seen from the positive values of Δ*f* in Table 6. Likewise, the results indicate that the arsenic atom is the site most prone to free radical attacks since it presents a matter of Δ*f* = −0.2156.

The results in Table 7 indicate that the TiCl_4_ chlorine atoms found at positions 2, 3, and 4 show greater susceptibility to electrophilic attack. On the other hand, titanium at position 1 is the prominent site for a nucleophilic attack. Regarding free radical attacks, it was also found that titanium in position 1 and chlorines in positions 2, 3, and 4 are the most susceptible sites (see Figure 6).

Regarding the support, it has been observed that certain atoms in positions 1, 3, 6, 9, 28, 30, and 37 are more prone to nucleophilic attack. These atoms are of particular interest because, according to existing research, alcohols tend to react with magnesium cations present at that position. To support these findings, data have been compiled in Table 8, which shows the specific values of *f*^0^, *f*^+^, *f*^−^, and Δ*f* for each of the atoms that make up the surface of 110 of the catalyst. These values provide evidence and support for the propensity of the mentioned atoms to be affected by nucleophilic attack.

### 3.3. Molecular Electrostatic Potential

Molecular electrostatic potential (MEP) is a fundamental tool used for analyzing and characterizing molecular structures and their chemical reactivity. This magnitude provides detailed information about the distribution of electric charges and the availability of electrons in a molecule, which is essential to identify and understand the nucleophilic (high electron density) and electrophilic (low electron density) regions [45]. The MEP allows us to visualize the areas of high electronic density, which are manifested as regions of lower electrostatic potential, and the areas of low electronic density, represented as regions of higher electrostatic potential within the molecule under study. In this way, the MEP provides a detailed picture of the electron density in different areas of each of the structures studied in this research, which is crucial to understanding their chemical reactivity. In addition, the MEP establishes correlations with other critical molecular properties, such as dipole moment, electronegativity, and atomic charges. These relationships extend our understanding of molecular polarity, electron affinity, and the chemical interactions that take place in the molecule, which are especially relevant for the prediction of chemical reactions and molecular behaviors [46,47].

In Figure 7a, you can see the graphical representations of the electrostatic potential in the propanol molecule. The regions of lower electrostatic potential, highlighted in intense red colors, correspond to highly reactive areas and are of great interest for possible electrophilic attacks. Specifically, these highly reactive regions are concentrated in the proximity of the oxygen atom in propanol, which suggests a greater availability of electrons in these areas, promising to be donated during the chemical reactions. In contrast, the areas of the highest electrostatic potential, visualized in striking blue tones, represent the most reactive sites for possible nucleophilic attacks on the propanol molecule. These highly nucleophilic regions are located mainly around the hydrogen atom attached to oxygen in propanol. This peculiar molecular arrangement suggests a higher affinity for receiving electrons in these areas, making them highly reactive sites for nucleophilic reactions. According to the analysis of the molecular electrostatic potential (MEP) map of the AsH_3_ molecule (Figure 7b), the sites most prone to undergo electrophilic and nucleophilic attacks are highlighted. The arsenic atom shows a region of intense red color in the MEP, indicating a higher electron density in its vicinity and, therefore, a deficiency of electrons in that area, which makes it highly electrophilic. This characteristic makes it a favorable target for reactions with electron-donating chemical species. On the other hand, the blue areas, located around the hydrogen atoms in the AsH_3_ molecule, exhibit a higher electron density, which makes them nucleophilic sites. These areas have a greater capacity to donate electrons and are conducive to interacting with electron-accepting chemical species. In Figure 7d, the MEP of TiCl_4_ is presented in a range that varies from −4.062 × 10^−4^ to 4.062 × 10^−4^ eV. The colors used in the MEP are blue, green, and red, representing the regions with the most positive electrostatic potential, zero potential, and negative electrostatic potential, respectively. It is observed that the blue regions, associated with the most positive electrostatic potential, are distributed around the entire TiCl_4_ molecule. This suggests a high probability that these regions are susceptible to nucleophilic attack due to their relatively low electron density.

On the other hand, in Figure 7c, the MEP of the MgCl_2_ support is shown in a range that goes from −4.564 eV to 4.564 eV. We observe that the green regions associated with the electrostatic potential close to zero are distributed around the entire molecule of the MgCl_2_ support. This suggests that these regions have a negligible probability of undergoing nucleophilic or electrophilic attack since they have a more balanced electron density.

### 3.4. Proposed Mechanisms of the Inhibition Process of Propanol and Arsine to the Ziegler–Natta Catalyst

The Ziegler–Natta catalyst uses a magnesium chloride (MgCl_2_) support essential for its catalytic function. When the oxygen atom of propanol interacts with a magnesium atom of the support, chemical complexes can be formed that affect the structure and stability of the catalyst, as shown in Figure 8a. This chemical interaction between oxygen and magnesium is due to the ability of oxygen to act as a ligand and coordinate with magnesium, which serves as a metal center on the MgCl_2_ support. Such an interaction has significant consequences on the performance of the catalytic converter. On the one hand, it can modify the selectivity of the catalyst in polymerization reactions. This can cause changes in the size distribution of the synthesized polymers or even lead to the production of undesired products, affecting the catalyst’s ability to insert the propylene monomer in a stereoselective manner, which would influence the regular arrangement of methyl groups in the polypropylene chain. If the stereoselectivity is negatively affected, the formation of isotactic polypropylene (with methyl groups in a regular arrangement) could decrease, resulting in a polymer with lower isotacticity and higher structure disorder [48]. Additionally, the formation of the complex between oxygen and magnesium can lead to a decrease in the activity of the main active site of the catalyst, which is TiCl_4_. In some cases, this interaction can completely inactivate the catalyst, affecting its ability to conduct polymerization reactions.

In Figure 8b, we can see that the first thing that happens is the adsorption of propanol on the surface of the TiCl_4_ catalyst, coordinating with the titanium atom, forming an intermediate species Ti-O-CH_2_CH_2_CH_3_. This prevents the TEAL co-catalyst from competing for active sites on the catalyst. The presence of propanol and the displacement of the ethyl group can cause alterations in the chemistry of the reaction, leading to changes in the speed of polymerization and the physical and mechanical properties of the resulting polymer. The interaction of methanol with the catalyst (TiCl_4_) can lead to a deactivation of the catalyst’s active site. When propanol interacts with the metal center of the Ziegler–Natta catalyst, it can form complexes that block or modify the active structure of the catalyst. This can significantly decrease its catalytic capacity and adversely affect the polymerization of the target olefins. Without propanol, the TEAL (trialkylaluminum) co-catalyst would combine with the catalyst to form a stable TEAL–TiCl_4_ complex. In this complex, the aluminum atom of TEAL donates a pair of electrons to the titanium atom of TiCl_4_, creating a coordinate bond between them. This binding of the TEAL to the catalyst activates the catalyst and facilitates the polymerization of olefins, such as propylene. The activated TEAL–TiCl_4_ complex can propagate the polymer chain, adding more propylene monomers and allowing the polymer to grow. The alkyl group of the TEAL binds to the active site of the catalyst, fulfilling a crucial role in stabilizing and activating the active centers of polymerization.

In the mechanism proposed in Figure 9, arsenic trifluoride (AsH_3_) competes with propylene monomer for the titanium (Ti) active site. First, a π complex is formed where AsH_3_ coordinates with the Ti of the TiCl_4_/MgCl_2_ complex. This interaction between AsH_3_ and Ti is carried out through the exchange of Ti, which is electronegative, with the pair of free electrons of arsenic, in which the electrons of the π bond of propylene predominate. The formation of the π complex in Ti-propylene has no barriers and occurs with a lower energy gain than the formation of the AsH_3_–Ti complex, so the latter reaction predominates. The insertion barrier of propylene varies between 6 and 12 kcal/mol. The high probability of the insertion reaction occurring is due to thermodynamics, given that a favoring of approximately 20 kcal/mol is observed.

In the polypropylene (PP) synthesis process, the polymer chain is propagated by moving the propylene towards Ti-PP, where the PP-alkyl chain is formed with the insertion of the olefin. PP chain growth is affected when the Ti-active center reacts with inhibitors of different polarities. In this case, AsH_3_ and propylene compete for the Ti-active site, which can affect the growth and propagation of the polypropylene chain during synthesis.

#### Interaction of Methanol, Arsine, and Isobutyl with the Ti-Active Center (Inhibitor–Ti Interaction)

It is important to clarify that we do not assert that the proposed adsorption models and mechanisms in this study correspond to actual active species. Instead, we believe that these models could serve as useful representations to investigate the adsorption and inhibition sites in ZN catalysts when present at low concentrations of detrimental substances during catalyst preparation or polymerization, as is the case with methanol and arsine.

In this section, we present the results obtained from simulating the binding of the two inhibitors to the TiCl_4_/MgCl_2_-active center. These calculations are based on the Mg_14_Cl_28_ cluster derived from relaxed MgCl_2_ surfaces, as shown in Figure 10. The selection of this model is grounded in previous calculations suggesting that the coordination of TiCl_4_ to the (104) plane is relatively weak or even unstable, in contrast to the energetically favorable coordination of TiCl_4_ to the (110) plane [49]. It has been verified that the Ti-isobutyl component, formed through 1,2-propylene insertion, constitutes the most fundamental unit that characterizes the growth of the polypropylene chain. The configuration of the uncoated active center TiCl_2_iBu/MgCl_2_(110) is displayed in Figure 10d.

The coordination of the two inhibitors with the titanium center results in values of −33.131, −60.824, and −46.171 kcal/mol for the Ti-isobutyl, AsH_3_, and CH_3_CH_2_CH_2_OH interactions, respectively. Indeed, the interaction between the inhibitors and the active Ti center emerges as the most favorable when compared to isobutyl, leading to the formation of a stable adduct complex (see Figure 11). To provide an insight into the stability of the poison-Ti adduct, the formation of a stable complex by complexing isobutyl at the same active site was examined. In fact, while the coordination of the monomer at the active Ti center exhibits favorable energetic stability, the complexes involving the analyzed inhibitors remain significantly stable. As a result, these inhibitors impede the formation of the alkene complex and the alkene insertion reaction. It is noteworthy to mention that this type of poisoning is nearly reversible: upon removal of the inhibitors from the system, the active centers resume the polymerization reaction [50].

### 3.5. Effects on the Melt Flow Index (MFI) and Mw of the PP

The behavior observed in the graph of the Melt Flow Index (MFI) versus molecular weight (Mw) of propylene with traces of propanol as an inhibitor reveals an interesting relationship between the concentration of propanol and the properties of the formed polymer (see Figure 12a). For insufficient concentrations of propanol (0 ppm to 11.33 ppm), the MFI remained constant at 20 g/10 min. The MFI measures the ease with which a molten polymer flows, and a constant MFI indicates that the polymer flow rate was not significantly affected by small amounts of propanol present at this stage [51]. However, from a propanol concentration of 11.33 ppm, the MFI began to gradually increase, reaching a value of 21.5 g/10 min at a concentration of 62.33 ppm. This suggests that at higher concentrations of propanol, the fluidity of the molten polymer increased, which could have implications for the processability and properties of the final product. On the other hand, the molecular weight (Mw) of polypropylene did not show drastic changes in the range of propanol concentrations. The Mw remained almost constant at around 36,000 kDa, with only slight variations observed, starting at 35.33 ppm and decreasing to 34,000 kDa. This indicates that propanol had a limited impact on the average molecular size of the polymer. Chemically, this observation suggests that small amounts of propanol may have affected the polymerization kinetics, influencing the reaction rate and, consequently, the MFI of the molten polymer. At low concentrations, propanol may have had an inhibitory action that kept the MFI constant. However, as the concentration of propanol increases, this inhibitor may begin to have a greater impact, as reflected in the gradual increase in the MFI. On the other hand, the limited impact on Mw suggests that small amounts of propanol may not have had a significant effect on the length of the polymer chains and, thus, on the average molecular size of the polymer.

The behavior of arsine is similar to that of propanol in the graph of the MFI (Melt Flow Index) versus Mw (molecular weight) of propylene. At low arsine concentrations (0 ppm to 0.06 ppm), the MFI remains constant at 20.0 g/10 min, indicating a stable molten fluidity at this stage. However, from a concentration of 0.06 ppm, the MFI begins to increase, reaching 21.5 g/10 min at a concentration of 0.035 ppm. Regarding the Mw of polypropylene, minimal changes are observed up to a concentration of 0.011 ppm of arsine, where the Mw is 39,000 kDa. The most pronounced difference is observed at a concentration of 0.035 ppm, where the Mw decreases at approximately 3600 kDa. The behavior observed with arsine suggests that, as with propanol, arsine may affect the polymerization kinetics. At low concentrations, arsine can act as an inhibitor that keeps the MFI constant. However, as the arsine concentration increases, this inhibitory effect decreases, leading to an exponential rise in the MFI. Regarding the Mw, the slight variations observed indicate that arsine may have a limited impact on the average molecular size of the polymer at low concentrations. However, at higher concentrations (0.035 ppm), a more significant decrease in Mw is observed, suggesting that arsine might have a more pronounced effect in controlling the polymer chain length at higher concentrations (see Figure 12b).

Both arsine and propanol show similar behaviors in the MFI, remaining constant at low concentrations and increasing at higher concentrations. However, in the case of propanol, the increase in the MFI starts from 11.33 ppm, while with arsine, it starts from 0.06 ppm. Regarding the Mw, propanol shows minimal changes in the range of concentrations, while arsine has a more significant decrease in Mw at 0.035 ppm. In both cases, the higher concentrations have a more substantial impact on the behavior of the MFI and the Mw of the polypropylene. The different forms of interaction of each compound with the catalyst could explain why similar behaviors are observed in the graphs (a constant MFI at low concentrations and an increase in the MFI at higher concentrations) but with different compound concentrations (propanol starting at 11.33 ppm; arsine starting at 0.06 ppm). Furthermore, the more pronounced decrease in Mw at 0.035 ppm arsine could result from its direct action on the active site of the catalyst, affecting the polymerization kinetics and polymer chain length.

### 3.6. Effects of Inhibitors on Catalyst Productivity

The analysis of the productivity graph of the Ziegler–Natta catalyst in different samples of polypropylene (PP) with varying concentrations of propanol and arsine reveals relevant information on the effect of these inhibitors on polymerization and catalytic efficiency. In the first sample of PP1 without propanol (Figure 13a), a catalyst productivity of 45 TM/kg was obtained. This indicates that the Ziegler–Natta catalyst performed well without propanol and achieved high polypropylene production. In sample PP2, where approximately five ppm propanol was added, the catalyst productivity remained similar to that of PP1. This suggests that, at this concentration, propanol did not have a significant impact on catalytic efficiency. However, as the concentration of propanol in the PP3 sample was increased (approximately seven ppm), a decrease in productivity was observed to 44 MT/kg. From 10 ppm, productivity continued to decrease, and at 52 ppm, the most significant productivity loss was obtained, with only 35 MT/kg. These results indicate that, at higher concentrations of propanol, the inhibitor began to impede catalyst activity, leading to a significant decrease in polypropylene production.

In the case of arsine (Figure 13b), changes in catalyst productivity were observed at lower concentrations than with propanol. Starting at approximately 0.006 ppm, productivity began to decrease. At a concentration of 0.024 ppm, productivity decreased to 39 MT/kg; at 0.0036 ppm, it further reduced to 36 MT/kg. These results suggest that propanol and arsine have different effects on the productivity of the Ziegler–Natta catalyst. With propanol, a loss of productivity is observed at higher concentrations, indicating that its interaction with the magnesium support or titanium-active site affects catalytic efficiency more significantly at higher concentrations. The decrease in productivity could be due to changes in the catalyst’s stability or alterations in the polymerization kinetics. On the other hand, with arsine, the decline in productivity is observed at much lower concentrations, suggesting that its direct action on the titanium-active site has a more rapid effect on catalytic efficiency.

In both cases, the specific interaction of each inhibitor with the catalyst and the changes in the reaction conditions could explain the observed differences in productivity. These results underline the importance of understanding how different inhibitors can affect polymerization and optimizing reaction conditions to obtain maximum efficiency in producing high-quality polypropylene.

### 3.7. Determination of the Isotacticity of PP with Traces of Propanol and Arsine

In the search for a deeper understanding of polypropylene (PP) behavior and its quality, a relevant solubility test in xylene has been carried out in the presence of traces of propanol and arsine. This evaluation provides valuable information on the interaction between PP and the compounds present and its influence on the polymer’s solubility. In addition, this study allows studying the possible effects in industrial and environmental applications of PP and offers a perspective to optimize processes and ensure the quality of the material. Likewise, the relevance of the isotacticity of PP is highlighted since its high stereoregularity directly impacts its mechanical and thermal properties. The presence of traces of propanol and arsine could alter the isotacticity index of PP, affecting its fundamental characteristics.

The results observed in Figure 14 revealed an interesting relationship between the concentrations of these compounds and the percentage of solubility of the PP in xylene. At low concentrations of propanol (0.5 to 15 ppm), a constant solubility of PP in xylene was observed at around 3.50%, indicating a good interaction between the polymer and the solvent. However, at higher concentrations of propanol (20 to almost 65 ppm), a gradual decrease in solubility was evidenced, suggesting competition for solvation sites in the PP. On the other hand, arsine showed a more pronounced effect at deficient concentrations (0.010 ppm or higher), with an abrupt reduction in the solubility of PP in xylene. These findings suggest that the presence of arsine significantly affects solubility, which could imply specific interactions with PP or xylene. These results provide valuable information on the solubility behavior of PP under different conditions and contribute to understanding the possible effects of these compounds on the industrial and environmental applications of the polymer.

## 4. Discussion

In this study, we have conducted research similar to that carried out by Bahri et al., focusing on the effect of propanol and arsine as inhibitors of the Ziegler–Natta catalyst, influencing the titanium-active site. We employed the DFT calculation method with the 6-311G basis set and the B3LYP functional method to calculate the adsorption energies. The obtained values for the adsorption energies of arsine, propanol, and isobutyl as a representative of propylene were −60.8 kcal/mol, −46.1 kcal/mol, and −33.1 kcal/mol, respectively. Our calculation methodology aligns with that of Bahri et al. in utilizing the density functional theory (DFT) and applying the B3LYP functional method to assess the interaction between poisons and the titanium-active site. However, it differs in the studied poisons and the obtained adsorption energies. Bahri et al. focused on poisons like Ti-H_2_O, H_2_S, CO_2_, O_2_, and CH_3_OH, evaluating their interactions with the titanium-active center on the MgCl_2_ surface and support [37]. Our results indicate that both arsine and propanol form stable complexes with the titanium-active center, suggesting potential catalytic poisoning. This complex stability aligns with the conclusions of Bahri et al., who also observed highly favorable poison-Ti interactions in their study. Furthermore, similarly to Bahri et al., we have found this type of poisoning to be largely reversible, as active centers regain their ability to carry out polymerization reactions when poisons are removed. Both Bahri et al.’s study and the current one underscore the importance of understanding the interaction between poisons and active centers in polymerization catalysts. Despite the use of different poisons and methodologies, the general conclusions point towards poisons’ capacity to influence catalytic activity and selectivity. Such research could contribute to strategies enhancing catalyst stability and efficiency in industrial applications. By comparing these two studies, it becomes apparent how distinct poisons and methodologies lead to comparable outcomes, reinforcing the robustness of the conclusions drawn. Collectively, these investigations advance the understanding and optimization of catalytic processes.

## 5. Conclusions

This study provides a detailed insight into the interactions between propanol and the Ziegler–Natta catalyst and the MgCl_2_ support. It was found that propanol coordinates on specific surfaces of the catalyst, where the magnesium atoms present vacant coordination sites, such as the (110) surface. The propanol interacts with the uncoated surface of the MgCl_2_ support via a Lewis base–Lewis acid interaction, where propanol binds to the surface oxygen atom and magnesium atom. When propanol interacts with the MgCl_2_ support, chemical complexes can form that affect the structure and stability of the catalyst. This interaction can modify the selectivity of the catalyst in polymerization reactions, which can result in changes in the size distribution of the synthesized polymers or even in the production of undesired products. In addition, the formation of the complex between the oxygen of the propanol and the magnesium of the support can decrease the activity of the main active site of the catalyst (TiCl_4_), even completely inactivating it in some cases. Theoretical calculations and analyses of frontier molecular orbitals and global descriptors revealed that propanol tends to donate electrons during chemical reactions, making it an electron donor in a reaction. On the other hand, arsine showed a high ability to donate electrons, while TiCl_4_ acts as an electron acceptor and exhibits more excellent stability due to its wide energy gap. In addition, the sites most prone to electrophilic, nucleophilic, and free radical attacks on propanol, MgCl_2_ catalyst, and support molecules were identified. These results indicate that certain propanol, arsine, and TiCl_4_ atoms are more susceptible to different types of chemical attacks.

When propanol is added to the TiCl_4_ catalyst, it is observed that an intermediate Ti-O-CH_2_CH_2_CH_3_ complex is formed, which prevents the TEAL co-catalyst from competing for active sites on the catalyst. This interaction can alter the chemistry of the reaction, affecting the rate of polymerization and the physical and mechanical properties of the resulting polymer. As for arsine, it can also compete with propylene monomer for the catalyst’s titanium (Ti)-active site. A π complex is formed where the arsine coordinates with the Ti of the TiCl_4_/MgCl_2_ complex. Arsine has a similar effect to propanol on polymerization, affecting catalyst productivity. However, arsine shows changes in productivity at much lower concentrations than propanol, suggesting that its action on the titanium-active site is faster. Regarding the effects on the Melt Flow Index (MFI) and the molecular weight (Mw) of polypropylene, it is observed that the MFI remains constant at low concentrations of propanol and arsine but increases at higher concentrations. This suggests that inhibitors can affect the fluidity of the molten polymer at different concentrations. However, the Mw of polypropylene shows minimal changes across the range of propanol and arsine concentrations, indicating that these inhibitors have a limited effect on the average molecular size of the polymer. In addition, a relationship is observed between the concentrations of propanol and arsine and the solubility of polypropylene in xylene. At higher concentrations of these inhibitors, the solubility of polypropylene in xylene decreases, suggesting that they may compete for solvation sites on the polymer. These findings are essential to understanding and optimizing polypropylene synthesis processes and guaranteeing the quality of the material obtained for various industrial and environmental applications.

## Figures and Tables

**Figure 1 polymers-15-03619-f001:**
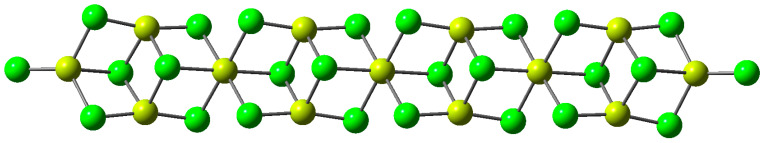
MgCl_2_ surface support 110.

**Figure 2 polymers-15-03619-f002:**
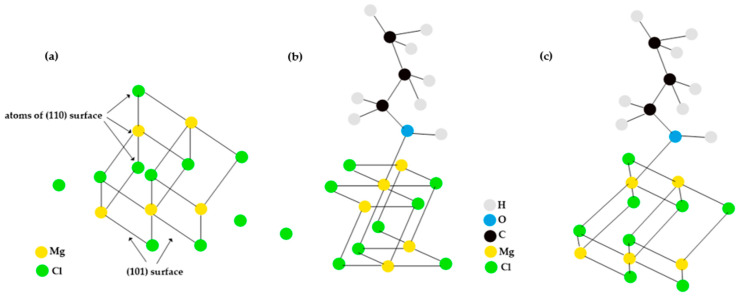
(**a**) Support model. (**b**) Coordination of propanol with the five-coordinate magnesium atom magnesium on the 101 surface. (**c**) Coordination of propanol with the five-coordinate magnesium atom magnesium on the 110 surface.

**Figure 3 polymers-15-03619-f003:**
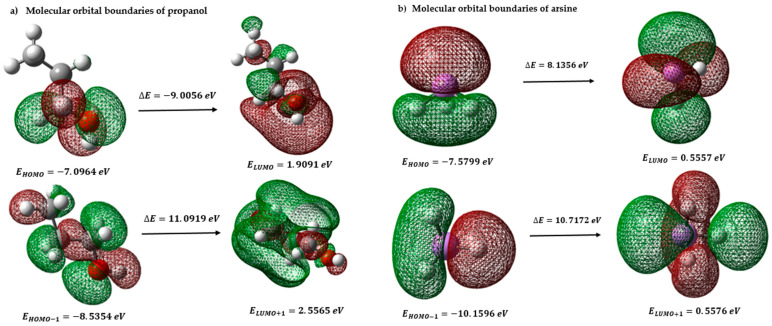
(**a**) Molecular orbital frontiers and possibilities of electronic transfer of propanol; (**b**) molecular orbital frontiers and possibilities of electronic transfer of arsine.

**Figure 4 polymers-15-03619-f004:**
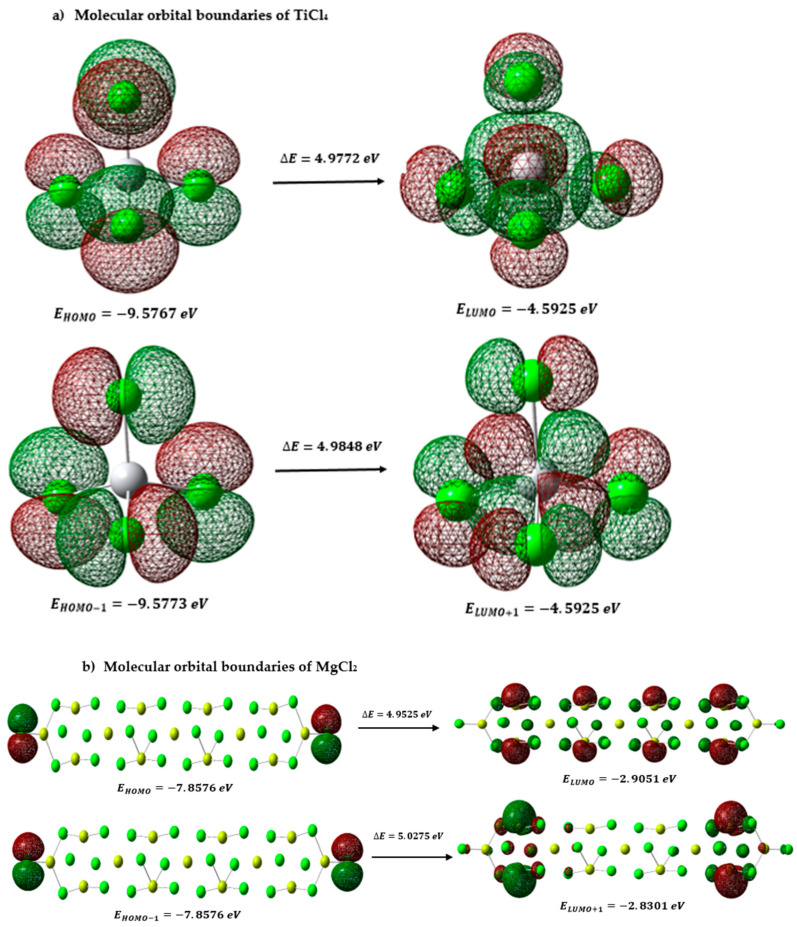
(**a**) TICL_4_ boundary orbitals, and (**b**) MgCl_2_ boundary orbitals.

**Figure 5 polymers-15-03619-f005:**
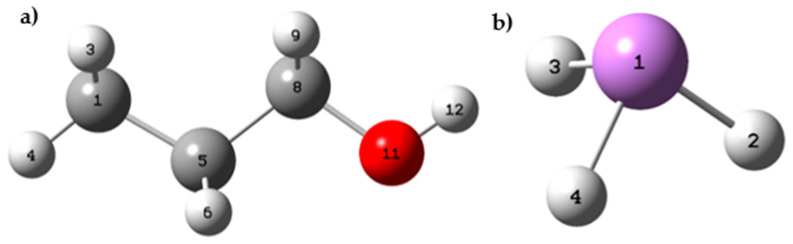
(**a**) Spatial conformation of propanol; (**b**) spatial conformation of AsH_3_.

**Figure 6 polymers-15-03619-f006:**
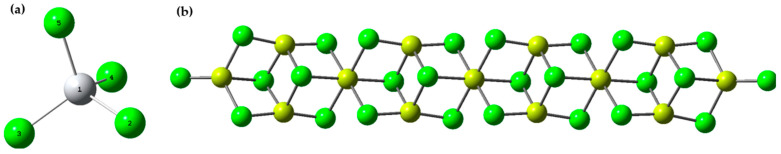
(**a**) Spatial conformation of TiCl_4_; (**b**) spatial conformation of MgCl_2_.

**Figure 7 polymers-15-03619-f007:**
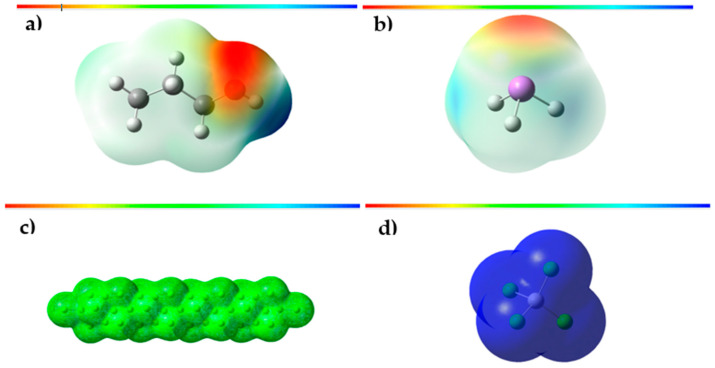
(**a**) Total density mapped with the electrostatic surface of propanol; (**b**) total density mapped with the electrostatic surface of arsine; (**c**) total density mapped with the electrostatic surface of the MgCl_2_ support; (**d**) total density mapped with the electrostatic surface of the TiCl_4_ support.

**Figure 8 polymers-15-03619-f008:**
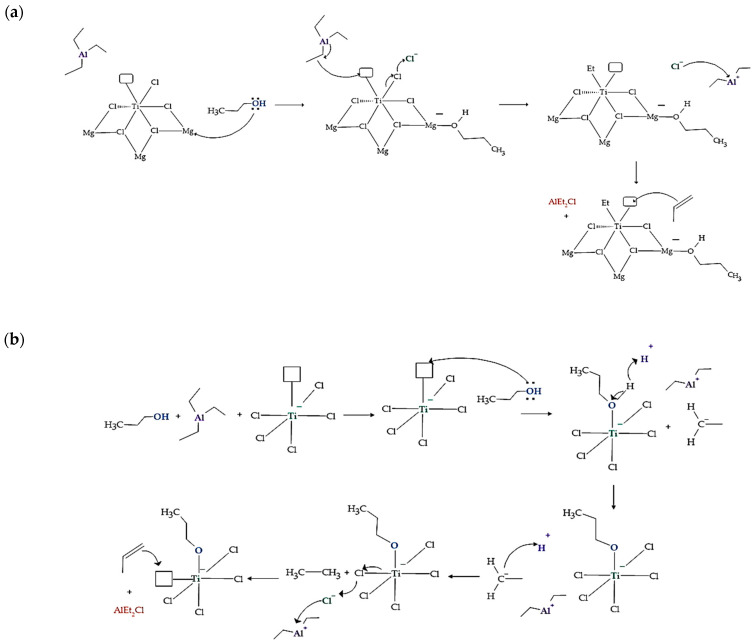
Proposed mechanisms for Ziegler–Natta catalyst inhibition: (**a**) propanol attacking the Mg of the support; (**b**) propanol acting on the active site of TiCl_4_.

**Figure 9 polymers-15-03619-f009:**
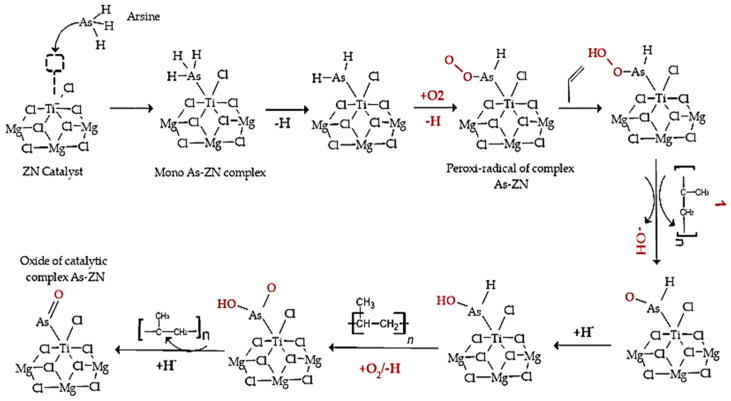
Proposed mechanisms for the inhibition of the Ziegler–Natta catalyst by arsine.

**Figure 10 polymers-15-03619-f010:**
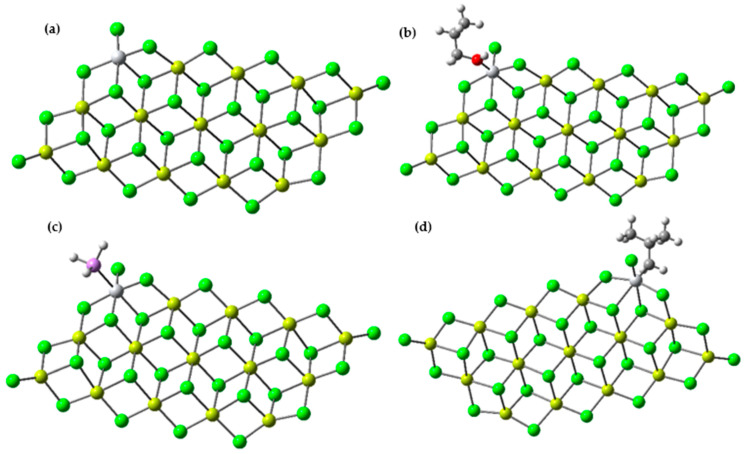
(**a**) Optimized structure of the TiCl_3_/Mg_14_Cl_28_ catalyst using the 6-311G base and the B3LYP-D3 functional; (**b**) optimized structure for the binding of CH_3_CH_2_CH_2_OH to Ti in the Mg_14_Cl_28_-TiCl_3_ model; (**c**) optimized structure for the binding of AsH_3_ to Ti in the Mg_14_Cl_28_-TiCl_3_ model; (**d**) optimized structure for the binding of isobutyl to Ti in the Mg_14_Cl_28_-TiCl_3_ model.

**Figure 11 polymers-15-03619-f011:**
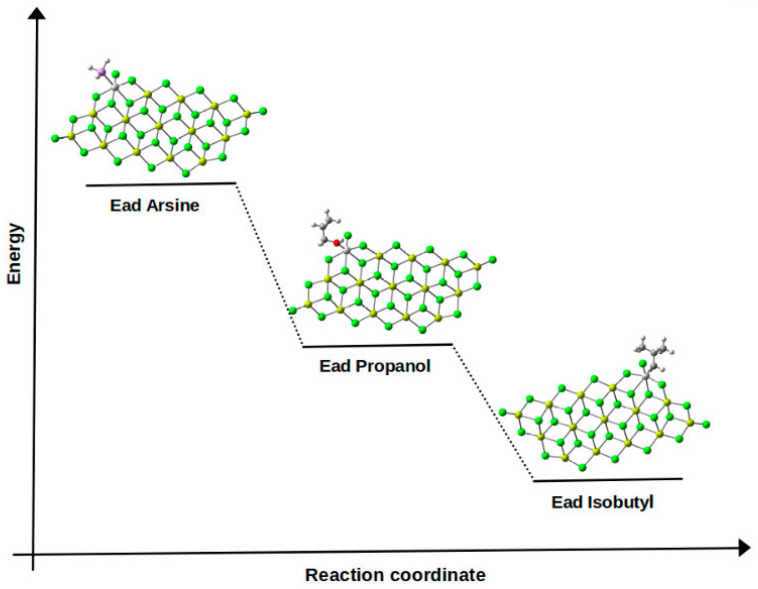
Diagram depicting the adsorption energies between inhibitors and the isobutyl.

**Figure 12 polymers-15-03619-f012:**
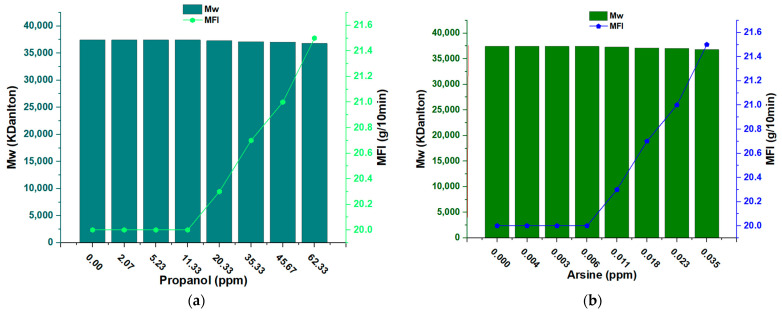
(**a**) Melt flow index and Mw of propanol; (**b**) melt flow index and Mw of arsine.

**Figure 13 polymers-15-03619-f013:**
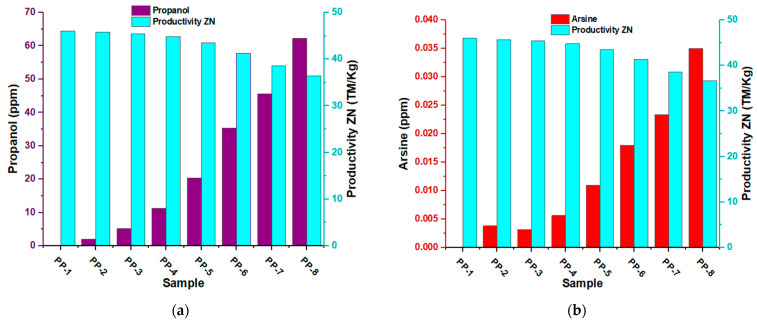
(**a**) Effect of propanol on the productivity of the Ziegler–Natta catalyst; (**b**) effects of arsine on the productivity of the Ziegler–Natta catalyst.

**Figure 14 polymers-15-03619-f014:**
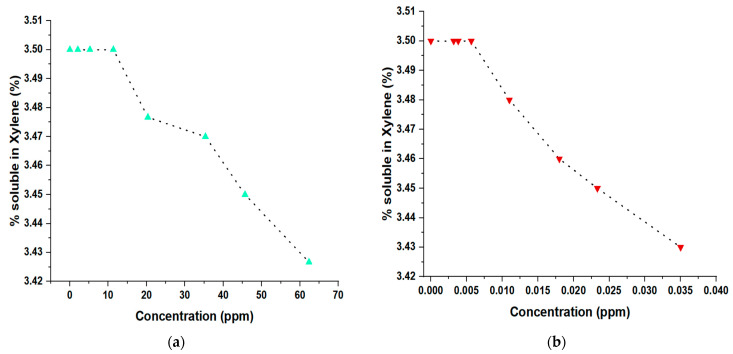
(**a**) Changes in the percentage of isotacticity of PP in the presence of propanol; (**b**) changes in the percentage of isotacticity of PP in the presence of arsine.

**Table 1 polymers-15-03619-t001:** Quantities of propanol collected during the polymerization process and measurements of MFI, Mw, and % soluble in xylene.

Propanol (ppm)	MT of PP Produced	Productivity Ziegler–Natta (TM/kg)	MFI	Soluble in Xylene (%Xs)
0	46	46	20	3.5
0	46	46	20	3.5
0	46	46	20	3.5
2.1	45.72	45.72	20	3.5
2	45.74	45.74	20	3.5
2.1	45.76	45.76	20	3.5
5.2	45.53	45.53	20	3.5
5.3	45.37	45.37	20	3.5
5.2	45.43	45.43	20	3.5
11	44.85	44.85	20	3.5
12	44.84	44.84	20	3.5
11	44.87	44.87	20	3.5
20	43.47	43.47	20.3	3.47
21	43.46	43.46	20.3	3.48
20	43.49	43.49	20.3	3.48
35	41.23	41.23	20.7	3.47
36	41.15	41.15	20.7	3.47
35	41.19	41.19	20.7	3.47
45	38.57	38.57	21	3.45
46	38.45	38.45	21	3.45
46	38.57	38.57	21	3.45
62	36.56	36.56	21.5	3.43
62	36.49	36.49	21.5	3.42
63	36.36	36.36	21.5	3.43

**Table 2 polymers-15-03619-t002:** Amounts of arsine collected during the polymerization process and measurements of MFI, Mw, and % soluble in xylene.

Arsine (ppm)	MT of PP Produced	Productivity Ziegler–Natta (TM/kg)	MFI	Soluble in Xylene (%Xs)
0	46	46	20	3.5
0	46	46	20	3.5
0	46	46	20	3.5
0.00095	45.68	45.68	20	3.5
0.00096	45.67	45.67	20	3.5
0.0096	45.69	45.69	20	3.5
0.0032	45.48	45.48	20	3.5
0.0033	45.47	45.47	20	3.5
0.0031	45.49	45.49	20	3.5
0.006	44.79	44.79	20	3.5
0.005	44.78	44.78	20	3.5
0.006	44.79	44.79	20	3.5
0.01	43.51	43.51	20.3	3.48
0.012	43.55	43.55	20.3	3.48
0.011	43.62	43.62	20.3	3.48
0.017	41.33	41.33	20.7	3.46
0.019	41.35	41.35	20.7	3.46
0.018	41.36	41.36	20.7	3.46
0.023	38.65	38.65	21	3.45
0.024	38.62	38.62	21	3.45
0.023	38.63	38.63	21	3.45
0.035	36.68	36.68	21.5	3.43
0.034	36.65	36.65	21.5	3.43
0.036	36.71	36.71	21.5	3.43

**Table 3 polymers-15-03619-t003:** Calculated energy values for propanol using B3LYP/6-31G.

Parameters	Energy Propanol	Energy Arsine
E_HOMO-1_ (eV)	−8.5354	−10.1596
E_HOMO_ (eV)	−7.0964	−7.5799
ΔE (eV)	9.0056	8.1356
E_LUMO_ (eV)	1.9091	0.5557
E_LUMO+1_ (eV)	2.5565	0.5576
η = ½ (E_LUMO_ − E_HOMO_) (eV)	4.5028	4.0678
χ = −1/2 (E_LUMO_ + E_HOMO_) (eV)	2.5936	3.5121
S = 1/η (eV)	0.2221	0.2458
µ = 1/2(E_LUMO_ + E_HOMO_) (eV)	−2.5936	−3.5121
ω = µ^2^/2η (eV)	0.7495	1.5162
N = E_HOMO(PR)_ − E_HOMO(TCE)_ (eV)	2.0645	1.581

**Table 4 polymers-15-03619-t004:** Calculated energy values for TiCl_4_ and MgCl_2_ using B3LYP/6-31G.

Parameters	Energy TiCl_4_	Energy MgCl_2_
E_HOMO-1_ (eV)	−95.773	−78.576
E_HOMO_ (eV)	−95.767	−78.576
ΔE (eV)	49.772	49.525
E_LUMO_ (eV)	−45.925	−29.051
E_LUMO+1_ (eV)	−45.925	−28.301
η = ½ (E_LUMO_ − E_HOMO_) (eV)	24.886	24.762
χ = −1/2 (E_LUMO_ + E_HOMO_) (eV)	70.846	53.813
S = 1/η (eV)	0.4018	0.4038
µ = ½ (E_LUMO_ + E_HOMO_) (eV)	−70.846	−53.813
ω = µ^2^/2η (eV)	100.843	51.161
N = _1_/ω (eV)	0.0992	0.1954

**Table 5 polymers-15-03619-t005:** Local descriptors of propanol.

Number	*f* ^+^	*f* ^−^	*f* ^0^	Δ*f*
1	0.0868	0.0019	0.0443	0.0849
2	0.0011	0.0001	0.0006	0.001
3	0.0011	0.0001	0.0006	0.001
4	0.0286	0	0.0143	0.0286
5	0.0229	0.0019	0.0124	0.021
6	0.0009	0.0102	0.0055	−0.0093
7	0.0009	0.0102	0.0055	−0.0093
8	0.2116	0.0317	0.1217	0.1798
9	0.0442	0.089	0.0666	−0.0448
10	0.0443	0.089	0.0666	−0.0447
11	0.4912	0.766	0.6286	−0.2748
12	0.0666	0	0.0333	0.0666

**Table 6 polymers-15-03619-t006:** Local descriptors of arsine.

Number	*f* ^+^	*f* ^−^	*f* ^0^	Δ*f*
1	0.6351	0.8507	0.7429	−0.2156
2	0.0232	0.0565	0.0399	−0.0334
3	0.2428	0.0565	0.1496	0.1862
4	0.099	0.0362	0.0676	0.0627

**Table 7 polymers-15-03619-t007:** Fukui functions for TiCl_4_.

Number	*f* ^+^	*f* ^−^	*f* ^0^	Δ*f*
1	0.7969	0.0001	0.3985	0.7969
2	0.0507	0.3742	0.2124	−0.3235
3	0.0504	0.164	0.1072	−0.1136
4	0.0512	0.374	0.2126	−0.3228
5	0.0508	0.0877	0.0693	−0.0369

**Table 8 polymers-15-03619-t008:** Fukui functions for MgCl_2_.

Number	*f* ^+^	*f* ^−^	*f* ^0^	Δ*f*
1	0.0807	0.0019	0.0413	0.0788
2	0.0166	0.0071	0.0119	0.0095
3	0.0581	0.0001	0.0291	0.058
4	0.0148	0	0.0074	0.0148
5	0.0174	0	0.0087	0.0174
6	0.0581	0	0.0291	0.0581
7	0.0139	0	0.007	0.0139
8	0.0139	0	0.007	0.0139
9	0.0807	0.0001	0.0404	0.0806
10	0.0174	0	0.0087	0.0174
11	0.0148	0	0.0074	0.0148
12	0.0166	0.0004	0.0085	0.0162
13	0.0027	0.9096	0.4561	−0.9069
14	0.0123	0.0183	0.0153	−0.0059
15	0.0095	0.0002	0.0049	0.0092
16	0.0068	0	0.0034	0.0068
17	0.0352	0.0006	0.0179	0.0346
18	0.008	0	0.004	0.008
19	0.0274	0	0.0137	0.0274
20	0.0087	0	0.0043	0.0087
21	0.008	0	0.004	0.008
22	0.0352	0	0.0176	0.0352
23	0.0068	0	0.0034	0.0068
24	0.0095	0	0.0047	0.0094
25	0.0123	0.001	0.0067	0.0113
26	0.0027	0.0507	0.0267	−0.048
27	0.0087	0	0.0043	0.0087
28	0.0807	0.0019	0.0413	0.0788
29	0.0166	0.0071	0.0119	0.0095
30	0.0581	0.0001	0.0291	0.058
31	0.0148	0	0.0074	0.0148
32	0.0174	0	0.0087	0.0174
33	0.0581	0	0.029	0.0581
34	0.0139	0	0.007	0.0139
35	0.0139	0	0.007	0.0139
36	0.0807	0.0001	0.0404	0.0806
37	0.0174	0	0.0087	0.0174
38	0.0148	0	0.0074	0.0148

## Data Availability

Not applicable.

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
