# Peer review of "Parts per Million of Propanol and Arsine as Responsible for the Poisoning of the Propylene Polymerization Reaction"

_polymers, 2023, doi:10.3390/polym15173619_

Round 1
Reviewer 1 Report
The manuscript by Fernández et al. examines the influence of propanol and arsine on the Ziegler-Natta catalyst in the polymerization of propylene. Despite the topic's potential significance, the experiments seem inadequately designed, which undermines the credibility of the manuscript's key conclusions. For this work to be considered for publication in "Polymers," several substantial issues need to be addressed:
- The authors' reliance on calculated properties like softness, nucleophilicity index, electrophilicity index, and hardness to suggest a new reaction mechanism is concerning. Such metrics are insufficient on their own. Energy profiles are needed to study the mechanism's thermodynamic and kinetic validity. Therefore, the authors should include energy calculations in their proposed mechanisms in both Figure 8 and Figure 9. Furthermore, propylene needs to be compared with propanol and arsine in these calculations to show the competing effect.
- The approach for determining the molecular weight of the resulting polypropylene, using the melt flow index with the Bremner approximation, is unusual. A more reliable technique, such as GPC, should be employed to ensure accuracy.
- Tables 1 and 2 are too crowded and difficult to interpret. For instance, in the row labeled “MYT of PP produced,” it’s hard to if the value of Run 4 is 45,7452 or 45,72. A restructuring of these tables, including the transposition of rows and columns and the replacement of commas with decimals, is essential for clarity.
- The claim on line 519, which suggests that increasing the concentration of propanol reduces its inhibiting effect, is counterintuitive and is presented without adequate explanation.
Author Response
Dear
We sincerely appreciate your detailed review and valuable suggestions regarding our manuscript titled "Parts per Million of Propanol and Arsine as Responsible for the Poisoning of Propylene Polymerization Reaction." Your careful analysis and insightful comments are of great assistance in enhancing the quality and contribution of our work.
We have taken into consideration your recommendations and, as a result, have carried out the restructuring of both tables, following each of your indications. Additionally, we have replaced commas with periods, as suggested, to enhance the clarity and readability of the presented data.
Furthermore, we have addressed your observation regarding the need to graphically present the interactions between propanol and the Ziegler-Natta catalyst, as well as with each of the poisons. Consequently, we have included a new figure that illustrates these interactions in a more visual and accessible manner. Additionally, we have calculated and compared the adsorption energies of each substance of interest in this study.
What we meant in line 519 is that as propanol concentration increases, it can start to have a greater impact on some physical properties of PP, as reflected by the gradual increase in MFI. This was corrected in the original text.
We greatly value your support and believe that your contributions have significantly enriched our manuscript. We extend our gratitude for your dedication and detailed attention. We hope that the modifications made have addressed your concerns and that our work meets the standards of quality and scientific rigor required for publication.
We remain at your disposal for any further comments or suggestions you may wish to share. We are committed to improving our work and contributing to the advancement of knowledge in this field.
We use the Bramner equation to predict the Mw. This equation uses the MFI as the calculation basis, and we have already used it in other research showing promising results. Unfortunately, in our laboratory, we do not have the GPC. It is super important to have this CPG since it would support our research very well. Unfortunately, the few economic resources available to our university do not allow us to buy a team like this. We also quoted another laboratory to carry out the tests, and they charged us very high economic amounts, and unfortunately we did not have these resources. These limitations have led us to use the Branmar equation as a support tool.
Kind Regards
Reviewer 2 Report
Review of the manuscript entitled ‘Parts per million of propanol and arsine as responsible for the poisoning of the propylene polymerization reaction’
Authors study propanol and arsine as inhibitors of the Ziegler-Natta catalyst in polypropylene production. They analyze how these two inhibitors affect the catalytic process and the properties of the resulting polymer is a novel and uncommon approach in the field. The calculations are done with the Gaussian 09 software and the theoretical method of the Density Functional Theory (DFT).
The study gives calculation results and compare with experimental results.
The interactions between propanol and the Ziegler-Natta catalyst and the MgCl2 support are given.
The manuscript is interesting and well written. It deserves to be published. The calculations associated with experimental results allow to understand and optimize polypropylene synthesis processes. With this study the quality of the material obtained for various industrial and environmental applications could potentially be optimize.
Author Response
Dear
Dear Reviewer,
We sincerely appreciate your thorough review and valuable feedback on our paper titled "Parts per Million of Propanol and Arsine as Responsible for the Poisoning of the Propylene Polymerization Reaction."
We are pleased that you have recognized the novel and uncommon approach of our study in analyzing the impact of propanol and arsine as inhibitors of the Ziegler-Natta catalyst in polypropylene production. Your observations highlight the importance of understanding how these inhibitors affect both the catalytic process and the properties of the resulting polymer, which opens new perspectives in the field of polymerization.
We are grateful for your acknowledgment that our manuscript is interesting and well-written, as well as for your support in considering it deserving of publication. Your evaluation underscores the relevance of our calculations in optimizing polypropylene synthesis processes, potentially making a significant impact on various industrial and environmental applications.
Thank you for your dedication and time spent reviewing our work. We remain at your disposal for any further inquiries or suggestions you may have. We hope that our article continues to meet the standards of quality and scientific rigor required for publication.
Kind Regards
Round 2
Reviewer 1 Report
In Figure 11, propilene should be propylene.
The adsorption energy -11366857.97 kcal/mol is unreasonably high. The authors should make sure that they have correctly subtracted the energies of the substrate and the adsorbent when calculating the adsorption energies.
Author Response
Dear
Thank you for evaluating this research. Now, we have made all the corrections.